# Prevalence and factors associated with female genital mutilation/cutting among Tanzanian women who gave birth in the five years prior to the survey: A population-based study

**Fabiola Vincent Moshi** ◉ *

Department of Clinical Nursing, School of Nursing and Public Health, The University of Dodoma, Dodoma, Tanzania

* fabiola.moshi@udom.ac.tz, fabiola.moshi@gmail.com

## Abstract

### Background

Female Genital Mutilation/Cutting (FGM/C) poses a significant public health challenge in developing countries, leading to increased risks of adverse obstetric outcomes such as caesarean section, postpartum hemorrhage, episiotomy, difficult labor, obstetric tears/lacerations, instrumental delivery, prolonged labor, and extended maternal hospital stays. The study aimed to determine the prevalence and factors associated with FGM/C among Tanzanian women who had given birth within five years preceding the Survey.

### Method

This study utilized an analytical cross-sectional design based on data from the 2015–2016 Tanzania Demographic and Health Survey and Malaria Indicators Survey (TDHS-MIS). A total of 5,777 women who had given birth within the five years preceding the survey and who provided responses to questions regarding female circumcision were included in the analysis. Descriptive analysis was employed to examine the prevalence of FGM/C among women in Tanzania. Additionally, multiple logistic regression was used to identify factors associated with FGM/C within this population.

### Results

The prevalence of FGM/C was 12.1% at 95%CI of 11.3% to 13%. Factors associated with FGM/C were marital status [married (AOR = 3.141 at 95%CI = 1.757–5.616,p<0.001), living with male partners (AOR = 2.001 at 95%CI = 1.082–3.699, p = 0.027), widowed (AOR = 2.922 at 95%CI = 1.201–7.111, p = 0.03)] never in union a reference population; wealth index [poorest (AOR = 2.329 at 95% CI = 1.442–3.763, p = 0.001), middle (AOR = 1.722 at 95% CI = 1.075–2.758, p = 0.024), richer (AOR = 1.831 at 95%CI = 1.205–2.781, p = 0.005)] in reference to richest women; zones [Northern zone, (AOR = 91.787 at 95%CI = 28.41–296.546, p<0.001), central zone, (AOR = 215.07 at 95%CI = 67.093–689.423, p<0.001), southern highlands, (AOR = 12.005 at 95% CI = 3.49–41.298, p<0.001), lake

**Data Availability Statement:** The datasets used during the current study is accessible through DHS Measure- TDHS-MIS 2015/2016 Survey Data

permission accessible at (https://dhsprogram.com/data/) Tanzania 2015-16 data set.

**Funding:** The author(s) received no specific funding for this work.

**Competing interests:** no competing interests exist.

**Abbreviations:** ANC, Antenatal Care; FGM/C, Female Genital Mutilation/Cutting; ICF, International's Institutional Review Board; NBS, National Bureau of Statistics; NIMR, Tanzania's National Institute for Medical Research; TDHS-MIS, Tanzania Demographic and Health Survey and Malaria Indicator Survey; USA, United States of America; WHO, World Health Organization; ZAMREC, Zanzibar Medical Ethics and Research Committee.

zone (AOR = 13.927 at 95%CI = 4.338–44.714,p<0.001), eastern zone, (AOR = 24.167 at 95% CI = 7.299–80.017, p<0.001)]; place of childbirth [outside health facility (AOR = 1.616 at 95%CI = 1.287–2.03, p<0.001)] in reference to health facility childbirth; parity [para 5+ (AOR = 2.204 at 95% CI = 1.477–3.288,p<0.001)] para one a reference population; and opinion on whether FGM/C stopped or continued [continued (AOR = 8.884 at 95% CI = 5.636–14.003, p<0.001).

## Conclusion

This study underscores the persistent issue of FGM/C in Tanzania, particularly among married women, those from lower-income households, and those living in regions with high prevalence. Women giving birth outside health facilities and those with multiple children are at higher risk. The study emphasizes the need for targeted interventions addressing socio-cultural factors, alongside providing legal, healthcare, and psychological support to those affected. Educational campaigns and community engagement, especially with traditional and religious leaders, are crucial for challenging cultural beliefs and reducing FGM/C's prevalence.

## Introduction

Female Genital Mutilation/Cutting (FGM/C) comprises of all procedures that includes the manipulation, alteration, or removing the external genital organs in young girls and women for non-medical reasons [1]. FGM/C is violation of human right of girls and women [1]. It reflects deep-rooted inequality between the sexes and constitutes an extreme form of discrimination against girls and women [1]. The procedure is performed using unsterilized tools such as blade or shard of glass by a religious leader, village elders, traditional female excisers or a medical professional with limited training [2]. FGM/C, as a violation of human rights, is often performed in secrecy, frequently under unsanitary conditions with unsterilized tools. This practice poses significant health risks, including infections like tetanus, sepsis, and urinary tract infections, which can result in severe, long-term health complications [1]. Moreover, because FGM/C is often conducted in secrecy by untrained practitioners, the procedure can result in severe complications such as excessive bleeding (hemorrhage), due to the use of sharp instruments or inadequate medical care. FGM/C performed in unhygienic conditions can have profound psychological effects, including anxiety, depression, and post-traumatic stress disorder (PTSD) [1]. The long term effects of the practice is an increased risk of caesarean section, postpartum hemorrhage, recourse to episiotomy, difficult labor, obstetric tears/lacerations, instrumental delivery, prolonged labor, and extended maternal hospital stay [3].

It is estimated that more than 200 million women and children have undergone FGM/C in 30 countries in Africa, the Middle East and Asia [4]. It is also estimated that about 3million girls are at risk of FGM/C each year globally [4]. The magnitude varies greatly between countries, regions and also within countries [5]. The most affected global regions are Africa and some Middle Eastern regions (including Iraq and Yemen) [5].

Tanzania is among the countries with a high prevalence of FGM/C. While the national prevalence stands at 10%, certain regions report alarmingly higher rates. The highest prevalence is in the Manyara region at 58%, followed by Dodoma at 47% [6]. Other regions with significant prevalence include Arusha (41%), Mara (32%), Singida (31%), and Tanga (14%).

These concerning statistics highlight the urgent need for targeted interventions to combat and eradicate FGM/C in the country [6].

According to World Health Organization (WHO) [1], FGM/C is classified into four main types depending on the action performed. Type one is when there is a partial or total removal of the clitoral glans (the external and visible part of the clitoris, which is a sensitive part of the female genitals), and/or the prepuce/clitoral hood (the fold of skin surrounding the clitoral glans). Type two is when there is a partial or total removal of the clitoral glans and the labia minora (the inner folds of the vulva), with or without removal of the labia majora (the outer folds of skin of the vulva). Type three is also known as infibulation, it is when there is a narrowing of the vaginal opening through the creation of a covering seal. The seal is formed by cutting and repositioning the labia minora, or labia majora, sometimes through stitching, with or without removal of the clitoral prepuce/clitoral hood and glans. Type four includes all other harmful procedures to the female genitalia for non-medical purposes, e.g., pricking, piercing, incising, scraping and cauterizing the genital area.

Women who experienced FGM/C can encounter physical health effects such as severe pain, bleeding, genital tissue swelling, and even death. The physical health effects can be a long-term complication including urinary problems, menstrual issues, painful intercourse, childbirth complications such as obstetric fistula, and increased risk of sexually transmitted infections such as HIV/AIDS [1]. FGM/C is also associated with a psychological and emotional effects. FGM/C is often performed without consent and can be traumatic, leading to long-term psychological effects such as anxiety, depression, and post-traumatic stress disorder. FGM/C as a part of psychological or emotional effect can negatively impact self-esteem and body image, leading to feelings of shame, inadequacy, and worthlessness. FGM/C can also lead to sexual problems such as pain during intercourse, decreased sexual satisfaction, and difficulty achieving orgasm.

FGM/C is associated with social effects such as stigma, social isolation and interpersonal relationship [1]. Tanzania legally prohibited FGM/C under its Sexual Offences Special Provision Act of 1998 [6]. The law states that anyone having custody, charge or care of a girl under eighteen years of age who causes her to undergo FGM/C commits the offence of cruelty to children [7]. The government has an obligation under international and regional human rights laws to ensure that the practice of FGM/C is eliminated [7]. The country has also adopted a National Plan of Action to end Violence against Women and Children and is committed to ending violence against women and children in all its forms, including FGM/C, by 2030 [6]. Despite of all these efforts, the FGM/C is still practiced. There exist variations in the country where the prevalence is high in Arusha, Dodoma, Manyara, Mara and Singida regions.

Having a law which enforce the sensation of the practice is an important step towards achieving women's sexual and reproductive health rights. In addition, a community sensitization on the impact of FGM/C is highly needed because many families allow FGM/C out of ignorance [8]. Families, communities and cultures in which FGM/C is performed, when asked why FGM is practiced they have different reasons for doing so. A major reported reason is that the practice is believed to ensure the girl conforms to key social norms, such as those related to sexual restraint, femininity, respectability and maturity [9]. All these can be achieved without subjecting women to FGM/C.

Understanding the prevalence and factors associated with FGM/C among women of reproductive age is crucial for improving childbirth complication readiness and addressing the specific factors contributing to this practice. Therefore, this study aimed at reporting the prevalence FGM/C among who had given birth within five years in Tanzania, along with the factors associated with the practice.

## Methods

### Study setting and design

Tanzania is located south of the equator, bordered by eight countries. Kenya and Uganda in the North, Rwanda, Burundi, the Democratic Republic of Congo, and Zambia in the West and Malawi and Mozambique in the South. The country is boarder by India ocean in the East. The country occupies an area of 945,087 km$^2$. The study design was an analytical cross-sectional study design using secondary data. The Tanzania Demographic and Health Survey (DHS) and Malaria Indicators Survey (MIS) 2015–2016 dataset was used to extract data to answer the research question. The survey was led by the National Bureau of Statistics (NBS) and the Office of Chief Government Statistician (OCGS), Zanzibar, in collaboration with the Ministry of Health in Tanzania Mainland and the Ministry of Health of Zanzibar.

### The 2015–16 TDHS-MIS

The 2015–16 Tanzania Demographic and Health Survey and Malaria Indicator Survey (2015–16 TDHS-MIS) is a nationally representative cross-sectional dataset. Its primary goal was to provide current estimates of key demographic and health indicators. The survey gathered data on various aspects, including fertility rates, marriage and sexual activity, fertility preferences, family planning awareness and usage, breastfeeding practices, nutrition, and maternal and child health. It also covered topics such as malaria and other health-related issues.

The sampling for the 2015–16 TDHS-MIS was conducted in two stages to ensure comprehensive coverage across Tanzania, including both urban and rural areas on the mainland and Zanzibar. The first stage involved selecting 608 clusters from enumeration areas defined during the 2012 Tanzania Population and Housing Census. In the second stage, 13,360 households were systematically chosen, of which 12,767 were occupied. Out of these, 12,563 households were successfully interviewed, resulting in a 98% response rate. Additionally, 13,634 eligible women were identified for individual interviews, with 13,266 completing the interview, yielding a 97% response rate.

### Study population, sample size and sampling

For this study, a subset of the original TDHS-MIS dataset was extracted based on specific criteria. The total sample size of women of reproductive age in the dataset were 13,266. Women of reproductive age who had given birth within five years preceding the survey were 6,924. Additionally, the outcome variable of interest (whether women had ever undergone female circumcision) was evaluated, and women with missing responses on this variable were excluded (1147 = 16.6%). Consequently, the final sample size used in the study comprised 5,777 women of reproductive age who gave birth five years preceding the survey.

### Data collection procedure

Both the household questionnaires and individual questionnaires were used for data collection. These tools were developed based on Measure DHS standards AIDS Indicator Survey and Malaria Indicator Survey questionnaires standards. The tools were translated into Kiswahili, the National language of Tanzania. A pre-test of the questionnaires was conducted in Tanga region from May 20, 2015, through June 18, 2015. Sixteen participants (12 women and 4 men) participated in the 4-week pre-test training and fieldwork practice for the 2015–16 TDHS-MIS.

Sixteen (16) field teams were used for data collection, three teams collected data in Zanzibar and 13 teams were allocated to collect data in Tanzania Mainland. Each team had four-wheel

drive vehicle with a driver, team supervisor, four female interviewers, one male interviewer, and one field editor, who also entered data into a tablet. The field editor and supervisor were responsible for reviewing all questionnaires for completeness, quality, and consistency before entering data into the tablet.

## Variables and variable measurements

**The independent variable.** Socio-demographic characteristics: All socio-demographic characteristics variables were categorical. These were, Age of woman with three categories (less than 20years, 20years to 34years and above 34 years), place of residence with two categories (urban and rural), marital status with six categories (never in union, married, living with partner, widowed, divorced and no longer living together/separated), wealth index with the five categories (poorest, poorer, middle, richer and richest) and zones of residence in Tanzania, categorized into nine zones (western zone, northern zone, central zone, southern highland zone, southern zone, south west highlands zone, lake zone, eastern zone and Zanzibar zone).

Obstetric Characteristics: All variables under obstetric characteristics were categorical variables. These were timing for initiation of antenatal clinics (ANC) which had two categories (early booking which was measured by first ANC visit within 12 weeks of gestation age and late booking measured by first ANC visit made after 12th week of gestation age). Number of ANC visits which was categorized into two categories, adequate visit and inadequate visit. Adequate was measured by having at least four ANC visits while inadequate visit was measured by making less than four ANC visits. Parity was categorized into three categories depending on the number of births (para one, para 2 to para 4 and para 5 and more). Place of childbirth had two categories, birth in health facility and birth outside health facility.

Opinion on female circumcision: this variable was measured by two categories depending on the opinion of interviewed women on whether female circumcision should be stopped or continued.

**Dependent variable.** FGM/C was the outcome variable, coded as 1 if a woman answered 'Yes' to having undergone female circumcision, and 0 if she responded 'No.' Those who answered 'Yes' were further asked three additional questions: 1) whether any flesh was removed from the genital area, 2) whether the genital area was merely nicked without removing any flesh, and 3) whether the genital area was sewn closed.

## Data analysis

Data were analyzed using Statistical Package for Social Sciences (SPSS) version 25. Descriptive analysis was done to analyze the characteristics of women of reproductive age to determine the frequency and percentages of each variable. To assess the factors associated with FGM/C, a cross tabulation was done between the independent variables (socio-demographic characteristics, obstetric characteristics and opinion on female circumcision) and dependent variable. All variables with p-value less than 0.2 were entered into regression model. Both Univariate and Multivariate logistic regression analysis was used to determine the strength of association between the independent and dependent variables. The variables with p-value less than 0.05 in the multivariate logistic regression were considered as factors associated with FGM/C.

## Ethics approval and consent to participate

This study analyzed secondary data (TDHS-MIS). No official ethical approval was needed. The permission to use the data for this particular research and publishing in peer reviewed journal was obtained from DHS measures. The procedures for collecting DHS-MIS data, however, were approved by the following organizations: Tanzania's National Institute for Medical

Research (NIMR), the Zanzibar Medical Ethics and Research Committee (ZAMREC), (ICF) International's Institutional Review Board, and the Centre for Disease Control and Prevention in Atlanta, USA. The participants' legal guardian/next of kin supplied written informed consent to participate in this study.

## Results

### Socio-demographic characteristics

A total of 5,777 women responded to the question whether they were circumcised or not. Majority of them were aged between 20 to 34 years (66.6%), were living in rural Tanzania (70.4%), had primary level of education (61.5%), were married (61.8%), had late antenatal booking (75.9%) (Table 1).

### Proportion of FGM/C among women of reproductive age in Tanzania

A total of 700(12.1%) at 95%CI of 11.3% to 13% of women of reproductive age were circumcised while a total of 5077(87.9%) of interviewed women were not circumcised (Fig 1).

Among 700 women who were circumcised, 616 (88%) had flesh removed from their genital areas, while 24(3.43%) no flesh was removed from their genital areas and 60(8.57%) did not know whether flesh was removed or not.

A total of 35(5%) women had their genital area sewn closed while 595(85%) their genital areas were not sewn closed and a total of 70(10%) did not know whether the genital areas were sewn closed or not.

A total of 172(3.0%) are in support that female circumcision has to be continued and a total of 5476 (94.8%) said female circumcision should be stopped while 48(0.8%) did not decide on whether to stop or continue and 81(1.4%) said they don't know (Fig 2).

### The relationship between women's characteristic and FGM/C status

Variables which showed significant relationship with FGM/C were age groups in years ($X^2$ = 52.765, p<0.001), type of place of residence (82.914 <0.001), highest education level ($X^2$ = 140.561, p<0.001), marital status ($X^2$ = 76.193, p<0.001), wealth index ($X^2$ = 275.829, p<0.001), zones ($X^2$ = 1487.434, p<0.001), parity of a woman ($X^2$ = 112.957, p<0.001) and place of childbirth ($X^2$ = 115.102, p<0.001) Table 2.

### Factors associated with female circumcision among women of reproductive age in Tanzania

After adjusted for confounders, factors associated with FGM/C among women of reproductive age in Tanzania were marital status [married (AOR = 3.141 at 95%CI = 1.757–5.616,p<0.001), living with male partners (AOR = 2.001at 95%CI = 1.082–3.699, p = 0.027), widowed (AOR = 2.922at 95%CI = 1.201–7.111, p = 0.03)] never in union was a reference population; wealth index [poorest (AOR = 2.329 at 95% CI = 1.442–3.763, p = 0.001), middle (AOR = 1.722 at 95% CI = 1.075–2.758, p = 0.024), richer (AOR = 1.831 at 95%CI = 1.205–2.781, p = 0.005)] in reference to richest women; zones [Northern zone, (AOR = 91.787 at 95% CI = 28.41–296.546, p<0.001), central zone, (AOR = 215.07 at 95%CI = 67.093–689.423, p<0.001), southern highlands, (AOR = 12.005 at 95% CI = 3.49–41.298, p<0.001), lake zone (AOR = 13.927 at 95%CI = 4.338–44.714,p<0.001), eastern zone, (AOR = 24.167 at 95% CI = 7.299–80.017, p<0.001)]; place of childbirth [outside health facility (AOR = 1.616 at 95% CI = 1.287–2.03, p<0.001)] in reference to health facility childbirth; parity [para 5+ (AOR = 2.204 at 95% CI = 1.477–3.288,p<0.001)] para one was the reference population; and

**Table 1. Socio demographic characteristics of women of reproductive age.**

| Times | Frequency (n) | Percent (%) |
|---|---|---|
| **Age Groups** | | |
| 15–19 years | 430 | 7.4 |
| 20 to 34 years | 3845 | 66.6 |
| 35–49 years | 1502 | 26 |
| **Type of place of residence** | | |
| Urban | 1709 | 29.6 |
| Rural | 4068 | 70.4 |
| **Highest educational level** | | |
| No education | 893 | 15.5 |
| Primary | 3552 | 61.5 |
| Secondary | 1273 | 22 |
| Higher | 59 | 1 |
| **Current marital status** | | |
| Never in union | 399 | 6.9 |
| Married | 3573 | 61.8 |
| Living with partner | 1099 | 19 |
| Widowed | 106 | 1.8 |
| Divorced | 315 | 5.5 |
| No longer living together/separated | 285 | 4.9 |
| **Wealth index** | | |
| Poorest | 1032 | 17.9 |
| Poorer | 1000 | 17.3 |
| Middle | 1098 | 19 |
| Richer | 1390 | 24.1 |
| Richest | 1257 | 21.8 |
| **Zones** | | |
| Western | 417 | 7.2 |
| Northern | 535 | 9.3 |
| Central | 675 | 11.7 |
| Southern Highlands | 484 | 8.4 |
| Southern | 295 | 5.1 |
| South West Highlands | 442 | 7.7 |
| Lake | 1438 | 24.9 |
| Eastern | 681 | 11.8 |
| Zanzibar | 810 | 14 |
| **Timing for ANC Booking** | | |
| Late booking | 4386 | 75.9 |
| Early booking | 1391 | 24.1 |
| **Number of ANC Visits** | | |
| Adequate | 3058 | 52.9 |
| Inadequate | 2719 | 47.1 |
| **Place of childbirth** | | |
| Health facility | 4008 | 69.4 |
| Outside health facility | 1769 | 30.6 |
| **Parity of the respondent** | | |
| Para one | 1395 | 24.1 |
| 2 to 4 | 2691 | 46.6 |
| Para 5+ | 1691 | 29.3 |

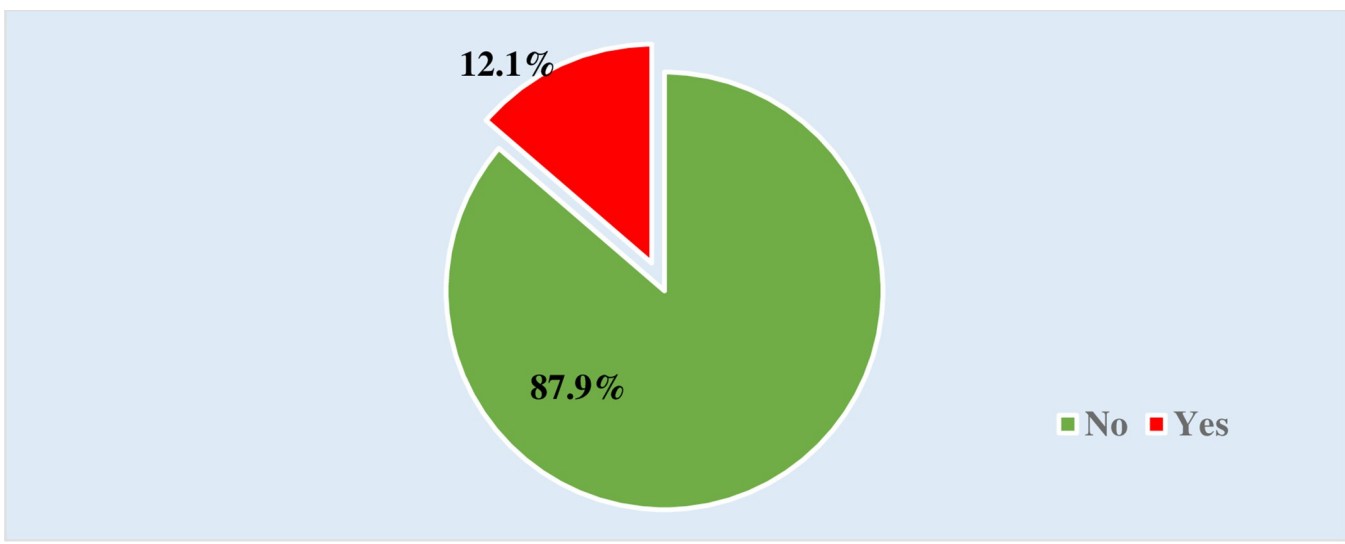

**Fig 1. Proportion of women of reproductive age who were circumcised.**

opinion on whether FGM/C stopped or continued [continued (AOR = 8.884 at 95% CI = 5.636–14.003, p<0.001)] (Table 3).

## Discussion

The findings from this study highlight that FGM/C remains a significant public health concern in Tanzania. Despite governmental efforts to criminalize the practice, the prevalence of FGM/C in the country remains high. The study revealed that 12.1% of women of reproductive age who had given birth five years preceding the survey in Tanzania have undergone FGM/C, highlighting the persistent nature of this harmful practice. Among these women, a significant

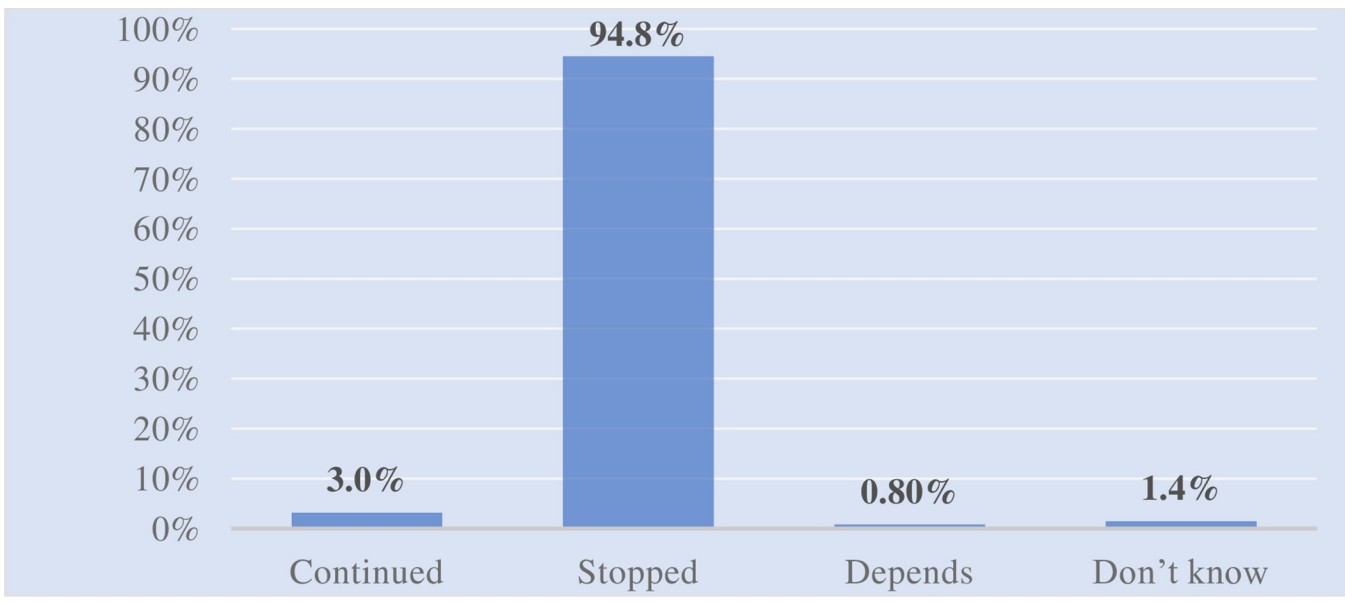

**Fig 2. Proportion of women in support of female circumcision be continued.**

**Table 2. The relationship between characteristics of women of reproductive age and FGM/C status.**

| Variable | Not Circumcised n(%) | Circumcised n(%) | $X^2$ | p-value |
|---|---|---|---|---|
| **Age Groups** | | | 52.765 | <0.001 |
| 15–19 years | 387(90) | 43(10) | | |
| 20 to 34 years | 3449(89.7) | 396(10.3) | | |
| 35–49 years | 1241(82.6) | 261(17.4) | | |
| **Type of place of residence** | | | 82.914 | <0.001 |
| Urban | 1605(93.9) | 104(6.10) | | |
| Rural | 3472(85.3) | 596(14.7) | | |
| **Highest educational level** | | | 140.561 | <0.001 |
| No education | 719(80.5) | 174(19.5) | | |
| Primary | 3075(86.6) | 477(13.4) | | |
| Secondary | 1225(96.2) | 48(3.8) | | |
| Higher | 58(98.3) | 1(1.7) | | |
| **Current marital status** | | | 76.193 | <0.001 |
| Never in union | 383(96) | 16(4) | | |
| Married | 3045(85.2) | 528(14.8) | | |
| Living with partner | 1002(91.2) | 97(8.8) | | |
| Widowed | 90(84.9) | 16(15.1) | | |
| Divorced | 298(94.6) | 17(5.4) | | |
| No longer living together/separated | 259(90.9) | 26(9.1) | | |
| **Wealth index** | | | 275.829 | <0.001 |
| Poorest | 770(74.6) | 262(25.4) | | |
| Poorer | 868(86.8) | 132(13.2) | | |
| Middle | 955(87.0) | 143(13) | | |
| Richer | 1272(91.5) | 118(8.5) | | |
| Richest | 1212(96.4) | 45(3.6) | | |
| **Zones** | | | 1487.434 | <0.001 |
| Western | 414(99.3) | 3(0.7) | | |
| Northern | 377(70,5) | 158(29.5) | | |
| Central | 323(47.9) | 352(52.1) | | |
| Southern Highlands | 461(95.2) | 23(4.8) | | |
| Southern | 291(98.6) | 4(1.4) | | |
| South West Highlands | 440(99.5) | 2(0.5) | | |
| Lake | 1327(92.3) | 111(7.7) | | |
| Eastern | 634(93.1) | 47(6.9) | | |
| Zanzibar | 810(1) | 0(0.0) | | |
| **Timing for ANC Booking** | | | 0.274 | 0.601 |
| Late booking | 3849(87.8) | 537(12.2) | | |
| Early booking | 1228(88.3) | 163(11.7) | | |
| **Number of ANC Visits** | | | 0.134 | 0.714 |
| Adequate | 2692(88) | 366()12( | | |
| Inadequate | 2385(87.7) | 334(12.3) | | |
| **Place of childbirth** | | | 115.102 | <0.001 |
| Health facility | 3645(90.9) | 363(9.1) | | |
| Outside health facility | 1432(80.9) | 337(19.1) | | |
| **Parity of the respondent** | | | 112.957 | <0.001 |
| Para one | 1292(92.6) | 103(7.4) | | |

*(Continued)*

**Table 2.** (Continued)

| Variable | Not Circumcised n(%) | Circumcised n(%) | $X^2$ | p-value |
|---|---|---|---|---|
| 2 to 4 | 2415(89.7) | 276(10.3) | | |
| Para 5+ | 1370(81) | 321(19) | | |
| Female circumcision: continue or be stopped | | | 335.897 | <0.001 |
| Stopped | 4891(89.30) | 585(10.70) | | |
| Continued | 74(43) | 98(57) | | |
| Depends | 41(85.40) | 7(14.60) | | |
| Don't know | 71(87.70) | 10(12.30) | | |

majority experienced severe forms of genital cutting: 88% reported that flesh was removed during the procedure, and 5% indicated that their genitals were sewn closed. These statistics highlight the severe extent of the harmful practices associated with FGM/C. In comparison, a systematic review and meta-analysis aimed at establishing the global prevalence of FGM/C among women of reproductive age reported a higher prevalence of 36.9%. The discrepancies in findings may be attributed to variations in study design and population [10].

A previous study assessing the prevalence of FGM/C in Africa and select Middle Eastern countries revealed a remarkable decline, particularly in East Africa. The most significant decrease was observed in East Africa, where the prevalence dropped dramatically from 71.4% in 1995 to just 8% in 2016 [5]. Despite this progress, Tanzania remains among the East African countries with a higher-than-average prevalence of FGM/C. This substantial reduction highlights encouraging progress in combating FGM/C in certain regions, underscoring the potential effectiveness of targeted interventions and advocacy efforts over time.

The persistent prevalence of FGM/C can be attributed to several factors. One significant reason is the widespread belief that the presence of the clitoris increases women's sexual desire, which is thought to lead to promiscuity and marital infidelity. This belief perpetuates the practice of FGM/C even in the face of legal prohibitions. Parents who subscribe to this belief seek out FGM/C services secretly, operating under strict secrecy to uphold these harmful cultural norms [8]. It is reported that majority of FGM/C inflicted upon girls between infancy and adolescence [11]. The minors are entirely dependent on their parents for protection and well-being. Research indicates that community sensitization and education are effective strategies for eradicating FGM/C [8]. Communities and tribal leaders need to understand the consequences of FGM/C and make informed choice.

There is compelling evidence demonstrating that FGM/C is not associated with increased promiscuity or marital infidelity [12]. Instead, research indicates that FGM/C is reported to significantly diminish the quality of sexual experiences. This includes increased sexual pain, difficulties with achieving orgasm, decreased sexual desire, challenges with arousal, and problems with lubrication. These negative impacts highlight the serious consequences of FGM/C on women's sexual health and well-being [13,14]. Additionally, FGM/C is linked to mental health disorders such as depression, somatization, anxiety, Post-Traumatic Stress Disorder (PTSD), and sleep disorders. These psychological consequences underscore the serious and lasting impact of FGM/C on women's mental well-being and overall health [13].

The study identified several factors associated with FGM/C, including marital status, wealth index status, geographical zones of residence, place of childbirth, parity, and women's opinions on whether the practice should be continued or stopped. These findings highlight the complex social, economic, and cultural factors that influence the prevalence of FGM/C within communities.

**Table 3. Factors associated with FGM/C among women of reproductive age in Tanzania.**

| Variables | COR | 95%CI | | p-value | AOR | 95%CI | | P-value |
|---|---|---|---|---|---|---|---|---|
| | | Lower | Upper | | | Lower | Upper | |
| **Age Groups** | | | | | | | | |
| 15–19 years | 1 | | | | 1 | | | |
| 20 to 34 years | 1.033 | 0.742 | 1.44 | 0.85 | 0.731 | 0.459 | 1.165 | 0.188 |
| 35-49years | 1.893 | 1.344 | 2.67 | <0.001 | 0.845 | 0.493 | 1.449 | 0.54 |
| **Type of place of residence** | | | | | | | | |
| Urban | 1 | | | | 1 | | | |
| Rural | 2.649 | 2.133 | 3.29 | <0.001 | 1.102 | 0.804 | 1.512 | 0.546 |
| **Highest educational level** | | | | | | | | |
| No education | 14.036 | 1.931 | 102 | 0.01 | 2.105 | 0.264 | 16.782 | 0.482 |
| Primary | 8.997 | 1.243 | 65.1 | 0.03 | 1.872 | 0.238 | 14.717 | 0.551 |
| Secondary | 2.273 | 0.308 | 16.8 | 0.42 | 1.4 | 0.176 | 11.13 | 0.75 |
| Higher | 1 | | | | 1 | | | |
| **Current marital status** | | | | | | | | |
| Never in union | 1 | | | | 1 | | | |
| Married | 4.151 | 2.496 | 6.9 | <0.001 | 3.141 | 1.757 | 5.616 | <0.001 |
| Living with partner | 2.317 | 1.348 | 3.98 | <0.001 | 2.001 | 1.082 | 3.699 | 0.027 |
| Widowed | 4.256 | 2.051 | 8.83 | <0.001 | 2.922 | 1.201 | 7.111 | 0.018 |
| Divorced | 1.366 | 0.679 | 2.75 | 0.38 | 1.202 | 0.537 | 2.687 | 0.655 |
| No longer living together/separated | 2.403 | 1.264 | 4.57 | 0.01 | 1.91 | 0.916 | 3.97 | 0.09 |
| **Wealth index** | | | | | | | | |
| Poorest | 9.164 | 6.596 | 12.7 | <0.001 | 2.329 | 1.442 | 3.763 | 0.001 |
| Poorer | 4.096 | 2.888 | 5.81 | <0.001 | 1.613 | 0.988 | 2.632 | 0.056 |
| Middle | 4.033 | 2.855 | 5.7 | <0.001 | 1.722 | 1.075 | 2.758 | 0.024 |
| Richer | 2.499 | 1.757 | 3.55 | <0.001 | 1.831 | 1.205 | 2.781 | 0.005 |
| Richest | 1 | | | | 1 | | | |
| **Zones** | | | | | | | | |
| Western | 1 | | | | 1 | | | |
| Northern | 57.836 | 18.3 | 183 | <0.001 | 91.787 | 28.41 | 296.546 | <0.001 |
| Central | 150.39 | 47.83 | 473 | <0.001 | 215.07 | 67.093 | 689.423 | <0.001 |
| Southern Highlands | 6.885 | 2.052 | 23.1 | <0.001 | 12.005 | 3.49 | 41.298 | <0.001 |
| Southern | 1.897 | 0.421 | 8.54 | 0.4 | 3.28 | 0.716 | 15.027 | 0.126 |
| South West Highlands | 0.627 | 0.104 | 3.77 | 0.61 | 0.92 | 0.151 | 5.605 | 0.928 |
| Lake | 11.543 | 3.647 | 36.5 | <0.001 | 13.927 | 4.338 | 44.714 | <0.001 |
| Eastern | 10.23 | 3.163 | 33.1 | <0.001 | 24.167 | 7.299 | 80.017 | <0.001 |
| Zanzibar | 0 | 0 | . | 0.99 | 0 | 0 | . | 0.991 |
| **Place of childbirth** | | | | | | | | |
| Health facility | 1 | | | | 1 | | | |
| Outside health facility | 2.363 | 2.013 | 2.77 | <0.001 | 1.616 | 1.287 | 2.03 | <0.001 |
| **Parity of the respondent** | | | | | | | | |
| Para one | 1 | | | | 1 | | | |
| 2 to 4 | 1.434 | 1.132 | 1.82 | <0.001 | 1.154 | 0.83 | 1.606 | 0.395 |
| Para 5+ | 2.939 | 2.324 | 3.72 | <0.001 | 2.204 | 1.477 | 3.288 | <0.001 |
| **Female circumcision: continue or be stopped** | | | | | | | | |
| Stopped | 1 | | | | 1 | | | |
| Continued | 11.072 | 8.09 | 15.153 | <0.001 | 8.884 | 5.636 | 14.003 | <0.001 |
| Depends | 1.427 | 0.637 | 3.196 | 0.387 | 1.497 | 0.552 | 4.06 | 0.428 |
| Don't know | 1.178 | 0.604 | 2.296 | 0.631 | 0.941 | 0.407 | 2.178 | 0.888 |

The study revealed significant variations in the prevalence of FGM/C across different zones of the country. More than half of the women interviewed in the central zone had undergone FGM/C. When compared to the western zone, the odds ratios were notably higher: 215 in the central zone, 92 in the northern zone, 24 in the eastern zone, 14 in the lake zone, and 12 in the southern highland zone. These zones represent different geographical areas where FGM/C occur varying proportions due to its deep cultural significance. The wide confidence intervals (CI) observed in the logistic regression model for different zones (Northern, Central, Southern Highland, Lake, Eastern) can be attributed to several factors such as variability in FGM/C prevalence, sample size variation or heterogeneity in risk factors across different zones of Tanzania FGM/C transcends being merely a traditional practice; it is deeply intertwined with notions of dignity, social status, and economic activity imposed by influential individuals within households and communities. This underscores the complex socio-cultural dynamics that perpetuate the practice and highlights the need for targeted interventions to address the root causes and drivers of FGM/C [6].

The study also revealed that marital status was a significant factor associated with FGM/C. Married women were three times as likely, those living with partners were twice as likely, and widowed women were almost three times as likely to have undergone female circumcision compared to women who had never been in a union. This increased likelihood could be attributed to the myth that circumcised women have reduced sexual desire, which may drive women to undergo FGM/C. Addressing this myth through community sensitization is essential, along with raising awareness about the harmful effects of FGM/C. Similar findings were reported by other studies investigating factors associated with FGM/C [15,16]. Different finding was reported by a similar study done in Chad [16]. The discrepancy in findings could be attributed to differences in how the marital status variable was categorized and analyzed across the two studies. In the previous study, marital status was simplified into two categories—either married or cohabiting. However, in this study, marital status was broken down into six distinct categories. Importantly, we used "never in union" as the reference category for comparison. This variation in categorization and analytical approach likely contributed to the contrasting results observed between the studies. The expanded categories in our study provide a more nuanced understanding of the relationship between marital statuses, highlighting the importance of considering such methodological differences when interpreting this outcome.

Furthermore, wealth index of a woman predicted the FGM/C status. Those with poorest wealth index were twice, those with middle wealth index were almost twice and those who were richer were 1.2 times more likely to undergo FGM/C than women who had richest wealth index. Similar studies have reported similar findings [16–18] A contradicting finding was reported by a study done in Ghana where wealth index was not a significant predictor of FGM/C [19]. The possible reasons could be due to differences in customs and traditions in these two countries.

Moreover, the study found the place of childbirth as a factor associated with FGM/C. Women who had their child born outside health facility were 1.6 times more likely to be circumcised. The possible explanation to this could be women who are circumcised may fear stigma among health workers and opt for home childbirth. Previous study have reported that women who are circumcised have fewer odds of using skilled birth attendants when compared with uncircumcised women [17,19]. The consequences of this is increasing more risk of birth complications by allowing birth under assistant of unskilled birth attendants.

In this study rural dwellings and low education attainment were not significant factors for FGM/C, they showed increased odds for the practice. A possible explanation for this finding is that FGM/C remains a deeply entrenched cultural practice, persisting irrespective of a woman's level of education or place of residence. In such contexts, cultural beliefs and social pressures often overpower the influence of education or urbanization.

The study also revealed that women with high parity were twice more likely to have FGM/C than those with low parity. High parity was a significant factor for FGM/C but this could be explained by fact that women with high parity were also those with low education, and living in rural settings. Previous studies have reported level of education and place of residence as factors associated with FGM/C [20].

Women who were in support FGM should continue were almost nine times more likely to be mutilated. Similar finding was reported by a previous study done in Nigeria [21]. It is evidenced that women who are circumcised are more likely to facilitate their baby girls to be circumcised [21]. This explains that changing women attitude towards FGM/C is a cornerstone strategy towards eradication of FGM/C. The findings of this study is published in research squire as a preprint [22].

While the study was robust in many aspects, its quantitative nature restricted the ability to provide narrative explanations of the findings. Furthermore, being a cross-sectional study may have limited the ability to establish a clear causal relationship among the reported factors. The reliance on self-reported data also presents the possibility of both over reporting and underreporting. Furthermore, the magnitude of FGM/C in this study might have been underestimated due to the exclusion of 16.6% of participants who did not respond to the question regarding their circumcision status".

## Conclusion

This study highlights that FGM/C remains a significant issue in Tanzania, affecting a notable portion of women. The analysis identified that women who are married, living with male partners, or widowed are at a higher risk of undergoing FGM/C compared to those who have never been in a union. Economic challenges also increase vulnerability, with women from lower-income households more likely to be subjected to the practice. Geographical disparities are evident, with Northern, Central, Southern Highlands, Lake and Eastern experiencing a much higher prevalence of FGM/C. The practice is more common among women who give birth outside health facilities and those with multiple children. Also, women who believe that FGM/C should continue are significantly more likely to experience it. To address this issue, it is essential to implement targeted interventions that focus on the socio-cultural factors sustaining FGM/C. Women at risk and those already affected should receive comprehensive support, including legal protection, healthcare, and psychological counseling. Educational campaigns should aim to challenge the cultural beliefs that perpetuate FGM/C, particularly in high-prevalence areas and among vulnerable groups, such as married women, those from low-income backgrounds, and those with larger families. Community engagement, especially with traditional and religious leaders, is crucial in shifting societal norms and reducing the practice's incidence.

## Supporting information

**S1 Checklist. STROBE statement—checklist of items that should be included in reports of *cross-sectional studies.***
(DOC)

## Acknowledgments

The author sends a sincere gratitude to DHS-Measure for allowing her to access the TDHS-MIS data set. The author also thanks Dr. Engelbert Bilashoboka for language editing and Dr. Bendera Anderson is acknowledged for reviewing the data analysis utilized in this study.

## Author Contributions

**Conceptualization:** Fabiola Vincent Moshi.

**Formal analysis:** Fabiola Vincent Moshi.

**Methodology:** Fabiola Vincent Moshi.

**Writing – original draft:** Fabiola Vincent Moshi.

**Writing – review & editing:** Fabiola Vincent Moshi.

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
