## [Decision Letter · Decision Letter 0]

19 Feb 2024

PONE-D-23-37734Determinants of Female Genital Mutilation/Cutting Among Women of Reproductive Age in Tanzania; an Analysis of the 2015–2016 Tanzania Demographic and Health Survey and Malaria Indicator Survey DataPLOS ONE

Dear Dr. Moshi,

Thank you for submitting your manuscript to PLOS ONE. After careful consideration, we feel that it has merit but does not fully meet PLOS ONE’s publication criteria as it currently stands. Therefore, we invite you to submit a revised version of the manuscript that addresses the points raised during the review process.

We look forward to receiving your revised manuscript.

Kind regards,

Joyce Jebet Cheptum

Academic Editor

PLOS ONE

2. PLOS requires an ORCID iD for the corresponding author in Editorial Manager on papers submitted after December 6th, 2016. Please ensure that you have an ORCID iD and that it is validated in Editorial Manager. To do this, go to ‘Update my Information’ (in the upper left-hand corner of the main menu), and click on the Fetch/Validate link next to the ORCID field. This will take you to the ORCID site and allow you to create a new iD or authenticate a pre-existing iD in Editorial Manager. Please see the following video for instructions on linking an ORCID iD to your Editorial Manager account: https://www.youtube.com/watch?v=_xcclfuvtxQ".

3. In the online submission form, you indicated that [These data are available upon request from DHS Measure- TDHS-MIS 2015/2016 Survey Data]. 

Additional Editor Comments:

This article needs to be reworked based on the comments as serious flaws in the methodology and results have been identified.

Reviewers' comments:

Reviewer's Responses to Questions

**Comments to the Author**

1. Is the manuscript technically sound, and do the data support the conclusions?

Reviewer #1: Partly

Reviewer #2: Partly

Reviewer #3: No

2. Has the statistical analysis been performed appropriately and rigorously? 

Reviewer #1: Yes

Reviewer #2: Yes

Reviewer #3: I Don't Know

3. Have the authors made all data underlying the findings in their manuscript fully available?

Reviewer #1: No

Reviewer #2: Yes

Reviewer #3: No

4. Is the manuscript presented in an intelligible fashion and written in standard English?

Reviewer #1: No

Reviewer #2: No

Reviewer #3: No

5. Review Comments to the Author

Reviewer #1: Title: it is confusing( somewhere it is determinant and somewhere it is prevalence), but in my reading I understood as it is a prevalence study, that's why you used a cross sectional study design,so please make your title as " prevalence and associated factors of FGM/C..."

Introduction section: the paragraphs need rearrangement, please keep the natural flow of ideas and your justification is not well stated, that is your interest of this study is not clear, why do you interested to conduct this study?

Your discussion is not presented in the intelligible way, please rewrite again

Your recommendation also poor, please write your recommendation based on your findings

You said "the study is without limitations......." but you put some limitations, what does that mean?

Reviewer #2: Thank you, dear editors, for you’re considering me to review this paper!

It is an interesting topic. However, I have the following comments to the authors for improvement:

Abstract:

Method: Do you appreciate the difference between a multivariate and a multivariable logistic regression? In the abstract section, you wrote multivariate logistic regression. Is multivariate logistic regression appropriate for your analysis?

Result: it is better to add the CI to AOR

Introduction

-It is good to refer to PLOS One as a guideline for reference citation. Your citation is superscripted.

Methods: you state that, ‘The study design was an analytical cross-sectional study design using secondary data and The Tanzania Demographic and Health Survey (DHS) and Malaria

The Indicators Survey (MIS) 2015–2016 dataset was used’.

-Why is pre-test? Since you used secondary data, why were sixteen (16) field teams used for data collection? Why Supervisor? Why interviewers?

-I expected the data to be extracted from the Tanzania Demographic and Health Survey (DHS) and Malaria Indicators Survey (MIS) 2015-2016 dataset as secondary data. But your data collection procedure describes how you collected primary data? Please try to describe the method clearly.

- Sample size calculation and sampling procedures are not clearly stated.

- NB: A more detailed method is needed to show how you extract your manuscript data from Tanzania Demographic and Health Survey (DHS) and Malaria Indicators Survey (MIS) 2015-2016 dataset.

Result:

- What is the importance of the relationship between the characteristics of women of reproductive Age and female genital Mutilation/Cutting Status? You have addressed the associated factors.

- How candidate variables selected for logistic regression analysis from a huge dataset?

-What are the reasons for the wide CI in Zones of Northern (28.41-296.546), Central (67.093- 689.423), Southern Highland (3.49 - 41.298), Lake (4.338 - 44.714) and Eastern (7.299- 80.017) in logistic regression model?

Reviewer #3: - There should be a clear objective for research

- What makes this research analytical? it is purely descriptive. if it is analytical cross sectional where is the hypothesis?

- There should be a clear methodology

- Should understand each components of research. the difference between result and discussion

6. PLOS authors have the option to publish the peer review history of their article (what does this mean?). If published, this will include your full peer review and any attached files.

Reviewer #1: No

Reviewer #2: No

Reviewer #3: No

---

## [Author Response · Author response to Decision Letter 0]

18 May 2024

Response to Reviewers’ Comments

Reviewer #1: 

Comment 1: Title: it is confusing (somewhere it is determinant and somewhere it is prevalence), but in my reading I understood as it is a prevalence study, that's why you used a cross sectional study design, so please make your title as " prevalence and associated factors of FGM/C..."

Response: Yes, the study used a cross-sectional design but it went further by controlling for possible confounders in regression analysis. Inferential statistics are used for analytical studies. Therefore, the study present both descriptive (prevalence) and analytical (determinants) findings.

Comment 2: Introduction section: the paragraphs need rearrangement, please keep the natural flow of ideas and your justification is not well stated, that is your interest of this study is not clear, why do you interested to conduct this study?

Response: The section is strengthened paragraph one and four

Comment 3: Your discussion is not presented in the intelligible way, please rewrite again

Response: The section is revised

Comment 4: Your recommendation also poor, please write your recommendation based on your findings

Response: The recommendations are revised

Comment 5: You said "the study is without limitations......." but you put some limitations, what does that mean?

Response: The language is revised; the author’s meaning was the study has limitation

Reviewer #2: 

Thank you, dear editors, for you’re considering me to review this paper!

It is an interesting topic. However, I have the following comments to the authors for improvement:

Comment 1: Abstract:

Method: Do you appreciate the difference between a multivariate and a multivariable logistic regression? In the abstract section, you wrote multivariate logistic regression. Is multivariate logistic regression appropriate for your analysis?

Response: The correction is made the study used multivariable logistics regression

Result: it is better to add the CI to AOR

Response: Confidence intervals were added

Comment 2: Introduction

-It is good to refer to PLOS One as a guideline for reference citation. Your citation is superscripted.

Response: The intext citation are revised to adhere to PLOS One instruction to authors

Methods: you state that, ‘The study design was an analytical cross-sectional study design using secondary data and The Tanzania Demographic and Health Survey (DHS) and Malaria

The Indicators Survey (MIS) 2015–2016 dataset was used’.

Response: The study was analytical because the analysis went further to controlling for confounders in the analysis. The regression analysis was used to assess the determinants of FGM/C

-Why is pre-test? Since you used secondary data, why were sixteen (16) field teams used for data collection? Why Supervisor? Why interviewers?

Response: The explanation in this section is according to the procedure taken by DHS Measure to ensure collection of credible and representative data, the section is revised to carry this meaning

-I expected the data to be extracted from the Tanzania Demographic and Health Survey (DHS) and Malaria Indicators Survey (MIS) 2015-2016 dataset as secondary data. But your data collection procedure describes how you collected primary data? Please try to describe the method clearly.

Response: The data were extracted from DHS-MIS, the procedure presented in the manuscript is according to what was done in the acquiring these secondary data

- Sample size calculation and sampling procedures are not clearly stated.

Response: The sample size based on the study criteria, women of reproductive age who responded to the question on whether they have undergone FGM/C were extracted from the total sample size and cleaning was done to omit all with missed data. The section is strengthened 

- NB: A more detailed method is needed to show how you extract your manuscript data from Tanzania Demographic and Health Survey (DHS) and Malaria Indicators Survey (MIS) 2015-2016 dataset.

Response: The section is revised

Comment 3: Result:

- What is the importance of the relationship between the characteristics of women of reproductive Age and female genital Mutilation/Cutting Status? You have addressed the associated factors.

Response: Relationship is the output of a cross tabulation analysis. It is a descriptive analysis which only describe the existence of a relationship between the independent variable and a dependent variable

- How candidate variables selected for logistic regression analysis from a huge dataset?

Response: All variables which showed a significant relationship in the cross-tabulation analysis were selected for logistic regression

-What are the reasons for the wide CI in Zones of Northern (28.41-296.546), Central (67.093- 689.423), Southern Highland (3.49 - 41.298), Lake (4.338 - 44.714) and Eastern (7.299- 80.017) in logistic regression model?

Response: A discussion to explain a possible reason for a wide confidence interval is added

Reviewer #3: - 

Comment 1: There should be a clear objective for research

- What makes this research analytical? it is purely descriptive. if it is analytical cross sectional where is the hypothesis?

- There should be a clear methodology

- Should understand each components of research. the difference between result and discussion

Response: 

Comment 2: It seems a review as it has critically evaluated the available data. It will help for a quick reference otherwise; the full version of the DHS has been available online for anybody. 

Response: It is true the DHS has a report but the report available is only descriptive. It reports about the prevalence of FGM/C. This study went further by conducting inferential analysis and report the factors associated with FGM/C. The findings from this study explain the characteristics of women of reproductive age who are likely to undergo FGM/C. This information is important for a well-directed effort on eradication the practice.

Comment 3: The data has been extracted from the DHS. My concern here is the data has been collected from permanently residents and visitors who stayed for one night, is it really make sense to collect a data from visitors especially on sensitive and culture related topics? Such kinds of practices are done secretly, how visitors will have the information?

Response: The individual file was used to extract data which was used in this study. Regarding to the sampling technique used by DHS-Measure, two clusters sampling techniques were used to sample the respondents, the first cluster sampling is selection of villages then the second cluster sampling was the selection of households, all women of reproductive age in the selected household were interviewed on the issue of female genital cutting. The data is a country representative data, the chance that the visitors may have diluted the picture of the FGM/C in Tanzania is minimal.

Comment 4: The other concern is In Tanzania the Government is working a lot to create awareness on FGM/C which has been a cause for the dropping of FGM/C from 10% (DHS 2015-16) to 8% (DHS 2022) as it has been justified in the report, there is a discrepancy between the main report and this research. The other point is this is purely cultural issue not related to knowledge and religion not only for Tanzania but also all over Africa. So, such kinds of sensitive issue it will be difficult to change the context only be awareness creation and the approach for the research should be different as it has to use behavioral and norm related model and theories to know the unwritten and unspoken hidden norms related with FGM/C, this will help all concerned bodies to critically plan to challenge the barriers. I do recommend PLOS ONE to critically evaluate papers for a real and meaningful contribution in research world, not for publication purpose but it has to be used as an input for the decision makers and other stake holders. 

Response: I do agree the practice of FGM/C is decreasing in the country due to the governmental and none governmental efforts. The current study did not study the trend in FGM/C but rather the determinants of FGM/C. The central focus of the current study is to understand the characteristics of women who undergo FGM/C so that to direct the efforts of the Government and non-governmental organizations. The magnitude reported in this study could have differed with the one reported in the DHS report because the current study included only women of reproductive age while the report included all women.

Comment 5: From my knowledge is concerned, in Africa the problem is no one is interested to explore a new knowledge but duplication. PLOSE ONE is a appreciated journal which challenges researchers than others, so avoiding such kinds of researches have to be avoided. There are lots of issues should be answered or little is known, the answer of this research topic is known “we have to avoid duplication”.

Response: I do agree with the reviewer that PLOSE ONE is a reputable journal and publish solid findings. It is with the same reason I chose to publish with them. The intention of this work is not to duplicate the available information but rather to add to what is already known. The DHS data reported the magnitude of FGM/C in the country and this study went further by reporting the characteristics of women who are more likely to be mutilated.

Comment 6: The most recent DHS result has been published; it is enough for reference and input purpose. Policy makers and other concerned bodies can refer the officially launched DHS report. For such kinds of culturally sensitive areas, we should use another innovative method to understand the deep-rooted causes to make a difference. For the publication purpose there are other journals open for any kinds of papers. 

Response: Findings from this study have additional information which can be used by policy makers to direct their decision making towards the practice

Comment 7: The result part is the main part of the paper which peoples quickly refer to understand the context but, in this paper, it has been written mainly in numbers (not interpreted in words) which can cause the readers a destruction and confusion. Not all people understand CI and P-value very well. The numbers should be explained in detail for anybody to understand what is really happening. 

Response: The numbers are interpreted in the discussion section 

Comment 8. The FGM/C especially in Tanzania is performed by traditional birth attendant and traditional circumciser because they share the same social norm with community they are serving. So whatever punishment and policy is applied, so what is the reason behind? Is the traditional way of research and method can really identify the factors and determinants? 

Response: FGM/C has impact in birth outcome. It is the role of the government to make the community aware of that. Punishment without sensitization and raising awareness is not correct. Punishment are used as a negative reinforcement of unwanted behavior. The study identified women’s characteristics which are likely to favor FGM/C

Comment 9: Most of the time reports related to FGM focus on the maternal health related consequences of FGM what about the violation of human right, mental health aspect, female sexual function (it is a deep-rooted problem for couples to divorce), infections and pains. Female has a right to enjoy her sexual life but the reality is different in a community which forces female to undergo this practice. I understood that the questioners are massive for DHS participants and self-administered, in Africa the literacy level is low to understand such kind of questions. And also, most of the time priority will be given for other indicators than such cultural but serious problems. Most indicators have been addressed by interview and supported by laboratory and physical examinations, when we come to the FGM/C case only by interview but at least it has to be triangulated with qualitative as the cases require a deep understanding by different approaches. The factors and determinants have not stated well or not clear, it has to be stated like among the determinants …. Factor is significantly associated with FGM/C or the like than numbers. Numbers can be summarized in table.

Response: The impact of FGM/C is expanded in the background information

Comment 10. N.B The paper lacks consistency, to be clearer. Nothing is said about the following issue under the result part but explained well under discussion. I think the researcher is not clear about what is the difference between result and Discussion. Some statements like a recommendation. 

Response: According to the instruction to authors, results present the output of analysis and the discussion serves the purpose of interpretation and discussion

---

## [Decision Letter · Decision Letter 1]

1 Aug 2024

PONE-D-23-37734R1Prevalence and Determinants of Female Genital Mutilation/Cutting Among Women of Reproductive Age in Tanzania; an Analysis of the 2015–2016 Tanzania Demographic and Health Survey and Malaria Indicator Survey DataPLOS ONE

Dear Dr. Moshi,

Thank you for submitting your manuscript to PLOS ONE. After careful consideration, we feel that it has merit but does not fully meet PLOS ONE’s publication criteria as it currently stands. Therefore, we invite you to submit a revised version of the manuscript that addresses the points raised during the review process.

We look forward to receiving your revised manuscript.

Kind regards,

Joyce Jebet Cheptum

Academic Editor

PLOS ONE

Journal Requirements:

Reviewers' comments:

Reviewer's Responses to Questions

**Comments to the Author**

1. If the authors have adequately addressed your comments raised in a previous round of review and you feel that this manuscript is now acceptable for publication, you may indicate that here to bypass the “Comments to the Author” section, enter your conflict of interest statement in the “Confidential to Editor” section, and submit your "Accept" recommendation.

Reviewer #2: All comments have been addressed

Reviewer #4: (No Response)

2. Is the manuscript technically sound, and do the data support the conclusions?

Reviewer #2: Yes

Reviewer #4: Yes

3. Has the statistical analysis been performed appropriately and rigorously? 

Reviewer #2: Yes

Reviewer #4: Yes

4. Have the authors made all data underlying the findings in their manuscript fully available?

Reviewer #2: Yes

Reviewer #4: Yes

5. Is the manuscript presented in an intelligible fashion and written in standard English?

Reviewer #2: Yes

Reviewer #4: No

6. Review Comments to the Author

Reviewer #2: (No Response)

Reviewer #4: Prevalence and Determinants of Female Genital Mutilation/Cutting Among Women of Reproductive Age in Tanzania; an Analysis of the 2015–2016 Tanzania Demographic and Health Survey and Malaria Indicator Survey Data.

This is an interesting paper that uses survey data to assess the prevalence of FGM and factors associate with the FGM. The paper continues to add to the body of knowledge on FGM. Although the paper provides useful information on the factors that are associated with FGM in Tanzania, the structure and the presentation in the manuscript may need some copy editing and formatting to ensure flow before publication. Detailed areas of recommendations are provided below.

Title: The title is well written providing information on the population and the study setting as well as the design employed. I however believe this could still be shortened as it seems too long. Alternative descriptive words could be used in the main body of the manuscript. Tanzania is repeated twice in the e title and could be omitted to shorten the title. Second, is the use of the term ‘determinants. This implies that the study is showing a causal relationship when in essence the study’s focus is to assess the factors associated with FGM.

Abstract: The abstract is well structured and concise. The word determinant and factors associated are used interchangeably in methods section when in deed they are distinct

Introduction: Generally, the introduction section is ok and provides the context of the study as well as the justification. There are a few areas that may need to be refined ---for example, the sentence that begin with, “The procedure is performed using unsterilized tools…..” seems to generalize that all FGMs and provides a biased analysis of the problem. The statement that begins with “improperly performed…” imply that there are proper ways of performing FGM.

There is a missing ‘to’ after according in the third paragraph. The word female genital mutilation/cutting has been abbreviated as FGM and can use the abbreviation thereafter. Otherwise the authors keep going back and forth with the use of the abbreviation and full description in the body of the manuscript.

Any reference to the high prevalence rates in the areas listed in Tanzania?

Methods: It would be helpful to provide a description of the TDHS-MIS and then the methods for this study. As it is, the methods being described in this study belongs to the TDHS-MIS. How was the data collected/abstracted for this study? What data cleaning procedures were conducted? How was missed data managed/handled? The data from women who were missing the information on whether or not they underwent FGM could bias your results – I can imagine the social desirability bias could have affected your findings here.

Why was the age categorized? For women aged less than 20 years, what was the lower age limit. Similarly, what was the upper age limit for those who were 34 and above. I want to believe the survey collects data from all women, and if this is not correct then your study background or the survey in which your study is based on, needs to be well described.

In the dependent variable section, delete the sentence that starts with ..Literature…

Which statistical analysis software was used to conduct the analysis?

Results: The results on circumcision is confusing. Were participants asked a follow up question on whether the flesh was removed or not? If that is the case, this is not provided in the methods section to understand this finding. Additionally, the finding with about 9% of the participants reporting not knowing whether or not the flesh was removed is more confusing – how did these persons know that they underwent FGM/C? Was there an examination of the genitalia to determine the responses?

Some figures could be left out as a sentence in a text communicated the information effectively, otherwise the figure just repeats what is in the text. If inserting figure 4, then there is no need of explaining everything in that figure within the text.

Discussion: This section is a little hard to follow. For example, paragraph 2is a little loaded with a lot of information that is hard to make out the implication of the study’s findings in comparison with what has been published out there. Paragraph 3 of the discussion does not discuss any of the study findings.

Paragraph 5 of the discussion does not seem to have provided a probable justification for the wide CI in the findings. What could have explained the study finding that education levels and rural dwelling were not significantly associated with FGM? Yet other studies have documented so?

Conclusion: The statement in the conclusion on the next step based on the findings should be focused. It is too general to relate to the findings of the study. What can be done about the women who have already undergone FGM and want this act to continue, hence the likely of them engaging this to their young girls?

7. PLOS authors have the option to publish the peer review history of their article (what does this mean?). If published, this will include your full peer review and any attached files.

Reviewer #2: No

Reviewer #4: No

---

## [Author Response · Author response to Decision Letter 1]

20 Aug 2024

Point to point response to reviewers’ Comments

General comment

This is an interesting paper that uses survey data to assess the prevalence of FGM and factors associate with the FGM. The paper continues to add to the body of knowledge on FGM. Although the paper provides useful information on the factors that are associated with FGM in Tanzania, the structure and the presentation in the manuscript may need some copy editing and formatting to ensure flow before publication. Detailed areas of recommendations are provided below

Response: Thank you

Comment 1.

Title: The title is well written providing information on the population and the study setting as well as the design employed. I however believe this could still be shortened as it seems too long. Alternative descriptive words could be used in the main body of the manuscript. Tanzania is repeated twice in the e title and could be omitted to shorten the title. Second, is the use of the term ‘determinants. This implies that the study is showing a causal relationship when in essence the study’s focus is to assess the factors associated with FGM.

Response: The title is shortened and the term determinants is replaced with factors

Comment 2

Abstract: The abstract is well structured and concise. The word determinant and factors associated are used interchangeably in methods section when in deed they are distinct

Response: The term determinants is replaced by factors throughout the document

Comment 3

Introduction: Generally, the introduction section is ok and provides the context of the study as well as the justification. There are a few areas that may need to be refined 

a. for example, the sentence that begin with, “The procedure is performed using unsterilized tools…..” seems to generalize that all FGMs and provides a biased analysis of the problem.

Response: The sentence is re-written and the bias in the sentence is addressed- line 87-90

b. The statement that begins with “improperly performed…” imply that there are proper ways of performing FGM.

Response: The sentence is re-written, focus is on surgical incision performed locally with untrained health care providers- line 92-98

c. There is a missing ‘to’ after according in the third paragraph. 

Response: The correction is made -113

d. The word female genital mutilation/cutting has been abbreviated as FGM and can use the abbreviation thereafter. Otherwise the authors keep going back and forth with the use of the abbreviation and full description in the body of the manuscript.

Response: The correction is made throughout the document

e. Any reference to the high prevalence rates in the areas listed in Tanzania?

Response: A paragraph is added which describe the regional disparities in FGM/C in Tanzania-line 107-112

Comment 4

Methods: 

a. It would be helpful to provide a description of the TDHS-MIS and then the methods for this study. As it is, the methods being described in this study belongs to the TDHS-MIS. How was the data collected/abstracted for this study? 

Response: The section has been revised in lines 171-185 to provide a clearer description of the method used in acquiring the DHS dataset.. 

b. What data cleaning procedures were conducted? 

Response: The cleaning procedure has been revised for clarity, as detailed in lines 193-199.

c. How was missed data managed/handled? The data from women who were missing the information on whether or not they underwent FGM could bias your results – I can imagine the social desirability bias could have affected your findings here.

Response: The proportion of missing data is now included in line 198. Respondents with missing data were excluded from the analysis. The authors acknowledge the reviewer's observation that this exclusion may have introduced bias into the findings, and this potential bias is addressed as a limitation in the study-line 444-448

d. Why was the age categorized? For women aged less than 20 years, what was the lower age limit. Similarly, what was the upper age limit for those who were 34 and above? 

Response: The objective was to analyze disparities in the prevalence of FGM/C across three distinct age categories of women of reproductive age: teenage mothers (15-19 years), middle-aged mothers (20-34 years), and older mothers (35-49 years). The age range for the third category has been revised to 35-49 years.

e. I want to believe the survey collects data from all women, and if this is not correct then your study background or the survey in which your study is based on, needs to be well described.

Response: The survey collected data exclusively from women of reproductive age. The aim of the current study has been modified in lines 144-149 to better reflect its focus on understanding the prevalence and contributing factors of FGM/C within this population.

f. In the dependent variable section, delete the sentence that starts with ...Literature…

Response: The sentence is deleted- line 236-239

g. Which statistical analysis software was used to conduct the analysis?

Response: The statistical analysis software used has been added to the Data Analysis section, specifically in line 246.

Comment 5

Results: 

a. The results on circumcision is confusing. Were participants asked a follow up question on whether the flesh was removed or not? If that is the case, this is not provided in the methods section to understand this finding. Additionally, the finding with about 9% of the participants reporting not knowing whether or not the flesh was removed is more confusing – how did these persons know that they underwent FGM/C? Was there an examination of the genitalia to determine the responses?

Response: The methods section has been enhanced, with a more detailed explanation of the dependent variable –line 236-243. However, it should be noted that the reliance on self-reported data may introduce bias into the findings. This potential limitation is addressed in the study's limitations section-

b. Some figures could be left out as a sentence in a text communicated the information effectively, otherwise the figure just repeats what is in the text. If inserting figure 4, then there is no need of explaining everything in that figure within the text.

Response: Agreed, figure 2 and 3 are removed

Comment 6

Discussion: secretly

a. This section is a little hard to follow. For example, paragraph 2is a little loaded with a lot of information that is hard to make out the implication of the study’s findings in comparison with what has been published out there. 

Response. The rearrangement of information is done between paragraph 1 and 2- line 330-337

b. Paragraph 3 of the discussion does not discuss any of the study findings.

Response: An introductory sentence is added to connect with previous presented results

c. Paragraph 5 of the discussion does not seem to have provided a probable justification for the wide CI in the findings. 

Response: The justification is clarifies-line 382-483

d. What could have explained the study finding that education levels and rural dwelling were not significantly associated with FGM? Yet other studies have documented so?

e. Response: A possible explanation is added -line 427-430

Conclusion: The statement in the conclusion on the next step based on the findings should be focused. It is too general to relate to the findings of the study. What can be done about the women who have already undergone FGM and want this act to continue, hence the likely of them engaging this to their young girls?

Response: The section is re-written

---

## [Editor Report · Decision Letter 2]

29 Aug 2024

Prevalence and Factors Associated with Female Genital Mutilation/Cutting among Tanzanian Women Who Gave Birth in the Five Years Prior to the Survey: A Population-Based Study

PONE-D-23-37734R2

Dear Dr. Moshi,

We’re pleased to inform you that your manuscript has been judged scientifically suitable for publication and will be formally accepted for publication once it meets all outstanding technical requirements.

Kind regards,

Joyce Jebet Cheptum

Academic Editor

PLOS ONE
---

## [Editor Report · Acceptance letter]

18 Sep 2024

PONE-D-23-37734R2 

PLOS ONE

Dear Dr. Moshi, 

I'm pleased to inform you that your manuscript has been deemed suitable for publication in PLOS ONE. Congratulations! Your manuscript is now being handed over to our production team.

Kind regards, 

on behalf of

Dr. Joyce Jebet Cheptum 

Academic Editor

PLOS ONE